



# Hydrological soil properties control tree regrowth after forest disturbance in the forest-steppe of central Mongolia

Florian Schneider[1], Michael Klinge[1], Jannik Brodthuhn[1], Tino Peplau[2], Daniela Sauer[1]

[1]Department of Physical Geography, University of Göttingen, Göttingen, 37077, Germany
[2]Thünen Institute of Climate-Smart Agriculture, Braunschweig, 38116, Germany

*Correspondence to*: Florian Schneider (florian.schneider@uni-goettingen.de)

**Abstract.** The central Mongolian forest-steppe is a sensitive ecotone, commonly affected by disturbances such as logging and

forest fires. In addition, intensified drought events aggravate stress on the trees that are anyway at their drier limit in the forest-steppe. Climate change increases evapotranspiration and reduces the distribution of discontinuous permafrost. The motivation for this study came about through our previous observation that forest stands show great differences with respect to their recovery after disturbance by fire or logging. Sometimes, no regrowth of trees takes place at all. As water availability is the main limiting factor of forest growth in this region, we hypothesized that differences in hydrological soil properties control the

forest-recovery pattern.

To test this hypothesis, we analysed soil properties under forests, predominantly consisting of Siberian larch (*Larix sibirica Ledeb.*), in the forest-steppe of the northern Khangai Mountains in central Mongolia. We distinguished four vegetation categories: 1. near-natural forest (FOR), 2. steppe close to the forest (STE), 3. disturbed forest with regrowth of trees (DWIR), and 4. disturbed forest showing no regrowth of trees (DNOR). 54 soil profiles were described in the field and sampled for soil

chemical, physical, and hydrological analysis. We found a significant difference in soil texture between soils under DWIR and DNOR. Sand generally dominated the soil texture, but soils under DWIR had more silt and clay compared to soils under DNOR. Soil pF curves showed that soils under DWIR had higher plant-available field capacity than soils under DNOR. In addition, hydraulic conductivity was higher in the uppermost horizons of soils under DWIR compared to soils under DNOR. Chemical properties of the soils under DWIR and DNOR showed no significant differences.

We conclude that the differences in post-disturbance tree regrowth are mainly caused by soil hydrological properties. High plant-available field capacity is the key factor for forest recovery under semi-arid conditions. High hydraulic conductivity in the uppermost soil horizons can further support tree regrowth, because it reduces evaporation loss and competition of larch saplings with grasses and herbs for water. Another important factor is human impact, particularly grazing on cleared forest sites, which often keeps seedlings from growing and thus inhibits forest recovery. Permafrost was absent at all studied disturbed

sites (DWIR, DNOR). We thus conclude that it is not a major factor for the post-disturbance tree-regrowth pattern, although it does contribute to water availability in summer.

**Keywords:** Siberian larch, permafrost, forest fire, human impact, climate change



## 1 Introduction

The forest-steppe in central Mongolia is an ecotone at the transition between the Siberian Taiga in the north and the Gobi Desert in the south, responding sensitively to climatic, ecological and anthropogenic disturbances. Water availability is the key factor that determines, where forest patches can exist within this ecotone. Climate change is a major threat for the forest-steppe due to aggravated drought stress (Allen et al., 2010). In Mongolia, intensified summer droughts became more frequent over the last century (Batima et al., 2005; Dashkhuu et al., 2015). Siberian larch (*Larix sibirica Ledeb.*), which is the dominant

tree species in the Mongolian forest-steppe, suffers from these intensified droughts (Dulamsuren et al., 2009; 2010; 2011; Chenlemuge et al., 2015b; 2015a). Furthermore, summer droughts increase the risk of fires that disturb the vegetation structure in forests and reduce the size of the forest patches (Kharuk et al., 2008). This causes a negative effect on the forest microclimate, which in turn intensifies the sensitivity of the forest to drought stress (Khansaritoreh et al., 2017a). In addition to fire, also human impact affects the structure of the forest stands. Logging and forest pasture reduce the forest size, open the forest stand

and hamper post-disturbance forest regrowth (Hilbig, 1987; Khishigjargal et al., 2013; Tsogtbaatar, 2013; Dulamsuren et al., 2014).

Meltwater from the active layer above permafrost patches increases the soil water availability and can thus support the survival of trees during drought events in summer (Sugimoto et al., 2002; Ishikawa et al., 2005; Churakova et al., 2016). Permafrost in Mongolia is discontinuous and is mainly limited to forests. However, climate change is currently reducing the area of

permafrost in Mongolia (Sharkhuu and Sharkhuu, 2012). After forest disturbance by fire or logging, permafrost degrades and may only reconstitute under certain conditions (Klinge et al., submitted). The loss of this additional water source leads to enhanced stress of trees during droughts.

Decreased water supply and increased drought stress, raises the importance of soil properties in this ecotone even further. Soil-water distribution in the landscape after a rainfall event is generally controlled by bedrock, sediment cover, slope morphology

and vegetation. In addition, plant-available field capacity and hydraulic conductivity of the soils influence the amount of water available at a site. Moreover, water repellency can reduce water infiltration, especially after fire events (DeBano, 2000; Doerr et al., 2000). Several studies have been conducted on soil distribution and soil properties in the Mongolian forest-steppe (Opp and Hilbig, 2003; Maximovich, 2004; Lehmkuhl et al., 2011). Nandintsetseg and Shinoda (2011) modelled soil moisture conditions in Mongolia based on data from meteorological stations from 1986 until 2005. They obtained a gradient of

increasing soil moisture from southwest to northeast during summer. Haase (1963) studied soils in relation to altitudinal zones in the Khangai Mountains, including particle size distribution, chemical soil properties, and water content. He reported forest vegetation on soils with higher water content and predominant steppe vegetation on soils with lower water content. However, he focused mainly on general soil description and did not report any further details on soil-vegetation relationships. Krasnoshchekov (2008, 2010) described physical and chemical soil properties in the East Khubsugul region and the Khangai

Mountains. Sympilova and Gyninova (2012) conducted similar research in the Selenga Mountains. However, in both studies only the soils were characterised, without linking them to the vegetation pattern. Sommer (2000) analysed soils of the forest-





steppe of the Turgen-Charchiraa-Mountains in north-western Mongolia. He focused on soil-chemical aspects and did not find a clear relationship between forest distribution and soil properties. He concluded that forest distribution was climate-controlled and disturbed by pastoral pressure and logging. Yet, he did not analyse soil-hydrological properties.

During previous fieldwork in the Khangai Mountains, in the central Mongolian forest-steppe, we observed that forest regrowth after disturbance does not proceed equally. The post-disturbance development of forest stands with apparently same site conditions ranges from spontaneous dense regrowth to no regrowth at all. As the trees are at their drier limit in this sensitive ecotone, we assumed that water availability is the main factor that controls forest regrowth after disturbance, whereas nutrient availability plays a less important role. Since both factors are related to soil properties, we established the following

hypotheses:

1) Silty soil texture leads to high plant-available field capacity and is thus favourable for post-disturbance tree regrowth.

2) High hydraulic conductivity in the uppermost parts of the soils supports post-disturbance forest regrowth, as rapid water infiltration to some depth reduces evaporation and ground-vegetation transpiration.

3) High nutrient stocks, for example after fire events, may support post-disturbance forest regrowth, unless nutrient stocks are
anyway well above the requirements of Siberian larch. In this case, no effect is to be expected.

4) Permafrost may increase water availability for trees through the continuous water release at the melting front above the permafrost table over the summer season. As water above the permafrost table cannot infiltrate downwards, interflow may occur above the permafrost table. This process leads to increased water availability, especially in concave slope positions where interflow converges. Thus, permafrost may support post-disturbance forest regrowth also at sites with otherwise
unfavourable soil-hydrological conditions for forest regrowth.

## 2 Materials and methods

### 2.1 Study area

The study area was located in the central Mongolian forest-steppe, in the northern Khangai Mountains, near the town Tosontsengel (Fig. 1). The geological basement predominantly consists of Permian acidic plutonic and metamorphic
sedimentary rock, with some occurrences of Carboniferous mafic rock (Academy of Sciences of Mongolia, Academy of Sciences of USSR, 1990). Slope debris with inmixed aeolian sand and silt provides the parent material for the soils in the region. The widespread aeolian deposits originate from lacustrine sediments of big lakes that dried out during the glacial periods in the Basin of Great Lakes, located between the Mongolian Altai and the Khangai Mountains (Grunert et al., 2000; Lehmkuhl and Haselein, 2000; Klinge and Lehmkuhl, 2013).







**Figure 1: Sentinel image (2019/07/31) of the study area with soil profile positions. The colours indicate the four vegetation categories that were used for selecting the soil profile locations: near-natural forest (FOR, green), steppe close to the forest (STE, grey), disturbed forest with regrowth of trees (DWIR, blue), and disturbed forest showing no regrowth of trees (DNOR, orange). The soil profiles are listed in Table 1 of the appendix. Right top: Physical map of Mongolia, showing the location of the study area (white rectangle).**


The climate is cold, semi-arid, and highly continental. In the town Tosontsengel, in the northern part of the study area, mean monthly temperatures range from -31.7 °C in January to 14.9 °C in July (Fig. 2). Mean annual precipitation is 200-500 mm. Rainfall concentrates in summer, because of low-pressure cells that are blown in with the westerlies (Batima et al., 2005). The

mean annual temperature of -5.9 °C supports discontinuous permafrost in the study area (Bonan and Shugart, 1989). Permafrost patches occur on forested slopes, where the canopy protects them from solar radiation, and the organic layer insulates them from high air temperatures in summer (Dashtseren et al., 2014). In summer, the forests benefit from an additional water supply by meltwater from the active layer above the permafrost (Zhang et al., 2011).





**Figure 2: Climate diagram for Tosontsengel, Mongolia (data source: meteorological station of Tosontsengel, Mongolia; National Statistical Office of Mongolia).**

The vegetation of the study area is dominated by steppe. Only north-facing slopes and some valley positions are covered by forest, predominantly consisting of Siberian larch (*Larix sibirica Ledeb.*) (Hilbig, 1987; Tsogtbaatar, 2013). The understory of these forests varies with respect to proportions of grasses, herbs, mosses and shrubs (commonly including e.g. *Vaccinium vitis-idea* and *Lonicera altaica*). High evapotranspiration combined with low precipitation and relief-dependant differences in insolation control the vegetation pattern (Schlütz et al., 2008; Hais et al., 2016). The semi-arid conditions promote frequent forest fires in this region (Goldammer, 2002; Hessl et al., 2016). The last two severe fire events in the study area happened in 1996 and 2002.

A timber factory was established in the 1960s in Tosontsengel. Thus, logging strongly affected the forest stands in the study area. Since the 1990s, industrial logging is abandoned, but illegal logging is still common (Lkhagvadorj et al., 2013). Moreover,





pastoral pressure increased over the last decades (Lkhagvadorj et al., 2013). Logging, followed by grazing, led to a reduction of the forested area, especially at its lower boundary and at the forest edges.

## 2.2 Field work

The site selection within the study area was based on four categories of vegetation: near-natural forest (FOR), steppe close to the forest (STE), disturbed forest with regrowth of trees (DWIR) and disturbed forest showing no regrowth of trees (DNOR) (Figs. 1 and 3). In addition, we distinguished three categories of disturbance intensity. "Low intensity" included e.g. logging of single trees. "Moderate" included logging of patches within a forest stand and fires that did not affect the whole forest stand. "Severe" included clear cutting and fires that destroyed the whole forest stand (Tab. A1, appendix). To reduce the effects of

factors that were not in the focus of this study, we kept geology, exposition (only north-facing slopes), inclination, and slope morphology as similar as possible. The elevation of the studied soil profiles ranged from 1850 to 2100 m a.s.l., which is well below the upper tree line of approx. 2500 m a.s.l. (Klinge et al., 2018). 54 soil profiles were described according to the FAO Guidelines for Soil Description (FAO, 2006) and classified according to WRB (IUSS Working Group WRB, 2015). The presence of permafrost was documented as well. The soil profiles were sampled horizon-wise, whereby thick horizons were

subdivided for the sampling, such that the maximum thickness included in one sample did not exceed 30 centimetres. We also carried out *in situ* measurements of saturated hydraulic conductivity, using a compact constant head permeameter (Amoozegar and Warrick, 1986; USDA, 2014). The measurements were done in five replicates. In addition, soil horizons with low rock-fragment contents were sampled with 100 cm³ and 250 cm³ steel cylinders for laboratory measurements of pF curves and hydraulic conductivity, respectively.



Near natural forest (FOR)

Steppe close to the forest (STE)

Disturbed forest with regrowth of trees (DWIR)

Disturbed forest showing no regrowth of trees (DNOR)




**Figure 3: Typical soil profiles under the four vegetation categories.**

## 2.3 Laboratory analyses

The soil samples were dried at 40 °C and passed through a 2 mm sieve. Approximately 10 g of each sample was dried at 105 °C to gravimetrically determine the remaining water contents for correcting all analytical data obtained from the 40 °C-dried samples. An aliquot of each sample was milled. Total C and N contents were analysed on the milled samples using a CHN analyser (LecoTruSpec). Carbonate contents were also analysed on the milled samples using the Scheibler method (Blume et al., 2011). Soil organic carbon (SOC) contents were calculated by subtracting soil inorganic C (obtained from carbonate analysis) from total C. Soil pH was measured in 1M KCl solution at a soil:solution ratio of 1:5. Exchangeable Al, Fe, Mn, Ca, Mg, K, and Na were extracted with 1 M ammonium chloride solution and measured by use of an ICP-OES (Thermo Scientific) (Lüer and Böhmer, 2000). The effective cation exchange capacity (ECEC) was calculated as the sum of negative charges occupied by these exchangeable cations. Pre-treatments for particle size distribution analysis were as follows: Soil organic matter was removed by 30 % $H_2O_2$, carbonates were dissolved, if necessary, by 10 % hydrochloric acid, and micro-aggregates were dispersed by 0.4 M sodium pyrophosphate (Deutsches Institut für Normung e.V., 2002; Durner and Iden, 2011). The three sand fractions were separated by sieving, and the silt and clay fractions were determined by use of a sedigraph (micromeritics). Bulk density was estimated in the field (FAO, 2006) and measured on undisturbed samples in the laboratory. All element concentrations in the fine earth were converted to element stocks per m² considering rock-fragment content and bulk density.

Hydraulic conductivity and pF curves were determined on 41 of the soil profiles under FOR, DWIR and DNOR. Hydraulic conductivity was measured in five replicates, using a permeability device with falling water head (Deutsches Institut für Normung e.V., 2012). Soil pF curves were determined in four to five replicates, using a pressure device (ecoTech) (Deutsches Institut für Normung e.V., 2014). Water repellency was analysed on five replicates for each top-soil sample (Doerr, 1998). The sample was placed in a petri dish and the surface was manually smoothened. The time was measured until a drop of distilled water, released from a pipette at 5 cm above the sample surface, infiltrated into the sample.

## 2.4 Data processing and statistical analysis

We described and sampled the soil profiles down to either the bedrock or permafrost, resulting in a maximum depth of 180 cm. According to the literature, the roots of Siberian larch can reach this depth (Kapper, 1954; Albenskiy et al., 1956). Therefore, we included all soil horizons in the calculations of element stocks to evaluate the nutrient availability for larch trees at a site. To assess in addition the site conditions also for tree seedlings without fully developed root system, we only used the properties of the uppermost 10 cm of the soils. For this purpose, we calculated the weighted means of the following variables for the uppermost 10 cm of the soils: content of rock fragments, bulk density, H+ concentration (log ($10^{-pH}$)), carbonate content, clay content, silt content, sand content, and ECEC (first group of variables). Element contents in the fine earth of each horizon were converted to element stocks per horizon (considering rock fragment content and bulk density), and were summed up over





the uppermost 10 cm of the soils. This was done for exchangeable Al, Ca, K, Na, Mg, Mn, Fe, SOC, and total N (TN) (second group of variables).

All statistical analyses were carried out with the R project for statistical computing (R Core Team, 2014). The share of each horizon in the uppermost 10 cm of the soil was used for the weighting, based on formula $x1$ (1) for the first group of variables, and formula $x2$ (2) for the second group of variables.

$x1 = if \ (upper \ boundary > 10) \{NA\} \ else \ \{$

$\quad\quad if \ (lower \ boundary <= 10) \ \{(lower \ boundary - upper \ boundary)/10\}$

$\quad\quad else \ \{(10 - upper \ boundary)/10\}\},$ $\hspace{4cm}$ (1)

$x2 = if \ (upper \ boundary > 10) \{NA\} \ else \ \{$

$\quad\quad if \ (lower \ boundary <= 10) \ \{1\}$

$\quad\quad else \ \{((lower \ boundary - upper \ boundary) - (lower \ boundary - 10))/(lower \ boundary -$

$upper \ boundary)\}\}$ $\hspace{6cm}$ (2)

We conducted all statistical analyses (i) on the entire data set of each profile, and (ii) on only the data of the uppermost 10 cm of each profile. All analyses and graphical representations were performed with the R packages "aqp", "ggplot2" and "ggpubr" (Beaudette et al., 2013; Wickham, 2016; Kassambara, 2019). Principle component analyses (PCAs) were calculated and graphically depicted with the R packages "FactoMineR" and "factoextra" (Lê et al., 2008; Kassambara and Mundt, 2019). The

PCAs were used to identify relations between the variables and the samples. The PCA of the uppermost 10 cm of the soils also included inclination, exposition and elevation. Linear regression models were used to compare the data obtained for soils under DWIR and DNOR. The results, including 95 % confidence intervals, described the differences between the soils under DWIR and DNOR.

## 3 Results

### 3.1 Field observations

The bedrock in the study area predominantly consisted of granite, with some minor occurrences of gneiss. Thick aeolian sand sheets locally covered the lower slopes and valley bottoms along west-east running valleys (Fig. 1). Thus, carbonates that were observed in some of the soils were supposed to originate from aeolian deposits.

The parent materials of the soils on the slopes generally consisted of a vertical succession of two to three sediment layers. The

lowermost one was typically a solifluction layer, characterised by a high content of rock fragments with sizes ranging from fine gravel to angular stones. This layer was usually overlain by another slope deposit, consisting of a mixture of weathering products of the bedrock, and aeolian sand and silt. This slope deposit showed lower rock-fragment contents than the underlying one, but similar rock-fragment sizes. The proportions of sand and silt in the upper slope deposit varied both within a profile





and between the profiles. In places, a Holocene colluvial deposit with varying thickness covered the slope deposits described
above. Our profiles 4, 5 and 38 predominantly consisted of Holocene colluvium, whereas most of the other profiles included
either no or only a thin Holocene colluvial deposit. The Holocene colluvial deposits had very low contents of rock fragments
that typically consisted of fine to medium gravel. At three sites, only a thin sediment layer covered the bedrock (profiles 3, 47,
52). As an exception, the parent material at one of the forest sites (profiles 1, 2, 7, 8, 9) consisted of a slope deposit with
extremely high proportions of aeolian sand.

Phaeozems and Cambisols were the most common soils under the near-natural forests (FOR) and disturbed areas (DWIR,
DNOR). In addition, Cryosols occurred above permafrost patches. Soils under steppe (STE) included Phaeozems, Chernozems,
and Kastanozems. The soil structure of the A horizons was generally granular, the B horizons had subangular or angular blocky
structure, and the C horizons had single grain, massive or rock structure. Rock fragments in several C horizons showed
manganese or carbonate pendants (Tab. A1, appendix). Bioturbation by rodents was observed in the soils at one forest site
(profiles 1, 2, 7, 8, 9), whereas cryoturbation was not found at any of the investigated sites, most likely because of insufficient
moisture. Charcoal was present in most of the topsoils.

Human impact affected most of the sites, except for the near-natural forest sites (FOR) that showed either no or only minor
disturbance such as cutting of single trees. Disturbance in DWIR and DNOR varied between moderate and severe. Most of the
sites were impacted by both fire and logging; yet, some profiles were only affected by one type of disturbance (Tab. A1,
appendix).

Permafrost was only encountered in soil profiles under near-natural forest (FOR). The depth of the permafrost table varied
between 60 cm and 100 cm in soils on slope debris under closed forests, and between 140 cm and 180 cm in soils on sandy
deposits under open forests (Tab. A1, appendix). In soils under STE, DWIR, and DNOR no permafrost was present above the
bedrock that was typically encountered between 80 cm and 110 cm soil depth.

## 3.2 Chemical and physical soil properties

Comparison of the entire soil profiles (down to either the bedrock or permafrost) showed that soils under STE had significantly
higher contents of carbonates and exchangeable Ca than soils under all other vegetation categories (Fig. 4), whereas soils under
FOR had significantly higher contents of exchangeable Fe compared to soils under all other vegetation categories. There were
no statistically significant differences between soils under DWIR and DNOR with respect to rock-fragment contents, SOC and
TN stocks, pH, contents of exchangeable Mg, K, Na, Al, Mn, and ECEC. However, soils under DWIR and DNOR differed
significantly in their particle size distribution: Soils under DWIR had higher silt and clay contents, whereas soils under DNOR
had higher sand contents. Soils under FOR had similar silt contents as soils under DWIR, whereas soils under STE had similar
sand contents as soils under DNOR. Bulk density of soils under FOR and DWIR was lower than that of soils under DNOR.
Yet, there was only a significant difference in bulk density between soils under FOR and DNOR but not between soils under
DWIR and DNOR.






**Figure 4: Properties of the entire soil profiles (down to either the bedrock or permafrost) under the four vegetation categories (n = 305): near-natural forest (FOR, green), steppe close to the forest (STE, grey), disturbed forest with regrowth of trees (DWIR, blue) and disturbed forest showing no regrowth of trees (DNOR, orange). P values indicated above the plots were calculated for differences between the arithmetic means (ns: p > 0.05; \*: p ≤ 0.05; \*\*: p ≤ 0.01; \*\*\*: p ≤ 0.001; \*\*\*\*: p ≤ 0.0001). Horizontal bars = medians, boxes = first and third quartiles, points = outliers, violins = data distributions.**

Comparison of only the uppermost 10 cm of each soil profile (for evaluating the soil conditions for tree seedlings) suggested that soils under DWIR had larger stocks of exchangeable Mg and K, SOC and TN than soils under DNOR, however without significant differences (Fig. 5). The other chemical soil properties showed no differences between soils under DWIR and DNOR. In contrast, particle size distribution showed again a significant difference between soils under DWIR and DNOR. Soils under STE had the highest sand contents, followed by soils under DNOR. Soils under DWIR had the highest clay and silt contents, followed by soils under FOR. Bulk densities and rock-fragment contents of the soils of the four vegetation categories were similar.











**Figure 5: Properties of the uppermost 10 cm of the soil profiles under the four vegetation categories (n = 54): near-natural forest (FOR, green), steppe close to the forest (STE, grey), disturbed forest with regrowth of trees (DWIR, blue) and disturbed forest showing no regrowth of trees (DNOR, orange). P values indicated above the plots were calculated for differences between the arithmetic means (NS.: p = 1; ns: p > 0.05; \*: p ≤ 0.05; \*\*: p ≤ 0.01; \*\*\*: p ≤ 0.001; \*\*\*\*: p ≤ 0.0001). Horizontal bars = medians,**
**boxes = first and third quartiles, points = outliers, violins = data distributions.**

In the principle component analysis (PCA) of the data for the entire soil profiles, the first two factors explained 56.3 % of the total variance (Fig. 6 top). The variables sand content, rock-fragment content and bulk density on one hand, made up one axis together with clay and silt contents on the other hand. Most of the data scattered around this axis, regardless of vegetation
category. Exchangeable Al and Fe on one hand, made up a second axis together with pH, carbonate content, and exchangeable Ca and Na on the other hand. Exchangeable Al and Fe did not contribute much to the PCA, yet, the soils under FOR clustered around them. On the other hand, some soils under STE, DNOR and FOR clustered around pH, carbonate content, and exchangeable Ca and Na. In addition, another cluster around SOC and TN stocks, and exchangeable Mg, K and Mn, occurred close to the axis of clay and silt contents. Mainly soils under DWIR but also some soils under STE fell into this cluster.

In the PCA of the data for the uppermost 10 cm of the soil profiles, the first two factors explained 54.5 % of the total variance (Fig. 6 bottom). Sand content and bulk density on one hand, made up one axis together with silt content on the other hand. Rock-fragment content and exchangeable Fe and Al on one hand, made up a second axis together with pH on the other hand. In addition, a cluster of clay content, SOC and TN stocks, ECEC, and exchangeable Mg, K and Mn occurred. The factor elevation also fell into this cluster, but with a minor contribution. The contributions of inclination, exposition, carbonate
content, and exchangeable Na were negligible. In general, the data for the uppermost 10 cm of the soils under the four vegetation categories (lower PCA) plotted more separately from each other than those for the entire soil profiles (upper PCA). The arithmetic means and 95 % confidence intervals of the data for the uppermost 10 cm of the soils under the four vegetation categories (smaller dots and shaded ellipses) confirmed the differences and clear separation. Soils under FOR plotted around the axes of sand and exchangeable Fe and Al. Soils under STE concentrated around sand and bulk density, thereby slightly
shifting to the side of exchangeable Ca, Mg, K, and SOC and TN stocks. Soils under DWIR plotted closer to silt than to sand, and mostly fell into the cluster of ECEC, exchangeable Mg, K, and SOC and TN stocks. Soils under DNOR plotted between those under STE and DWIR. The confidence interval of the soils under FOR was clearly separated from all other confidence intervals. That of the soils under DWIR was also separated and showed only a minor overlap with the one of soils under DNOR. The confidence intervals of the soils under STE and DNOR showed a considerable overlap.






**Figure 6: Top: PCA of the properties of the entire soil profiles under the four vegetation categories (n = 303): near-natural forest (FOR, green), steppe close to the forest (STE, grey), disturbed forest with regrowth of trees (DWIR, blue) and disturbed forest showing no regrowth of trees (DNOR, orange). Bottom: PCA of the properties of the uppermost 10 cm of the soil profiles under the four vegetation categories (n = 52). In addition, inclination, exposition and elevation were included in the second PCA. Arithmetic means and 95 % confidence intervals of the data for soils under each vegetation category are represented by a smaller circle and a shaded ellipse, respectively. SOC = soil organic carbon, TN = total nitrogen, ECEC = effective cation exchange capacity, BD = bulk density, and Ca, Mg, K, Na, Al, Fe, and Mn = exchangeable cations.**

Based on the data presented in Figures 4-6, we chose the most relevant variables for direct comparison of the soil properties under DWIR and DNOR, using multiple linear regression models (Fig. 7). We found a significant difference in particle size distribution, both for the data of the entire soil profiles (Fig. 7 top) and for the uppermost 10 cm of the soil profiles (Fig 7. bottom). Soils under DNOR had clearly more sand, and less silt and clay than soils under DWIR. The soils did not exhibit significant differences with respect to any other soil properties. All confidence intervals overlapped with the zero line, and all p values were >0.05. The only relevant observation was that the uppermost 10 cm of the soil profiles under DNOR tended to have smaller stocks of SOC and exchangeable K, compared to those under DWIR, however without significant differences either.

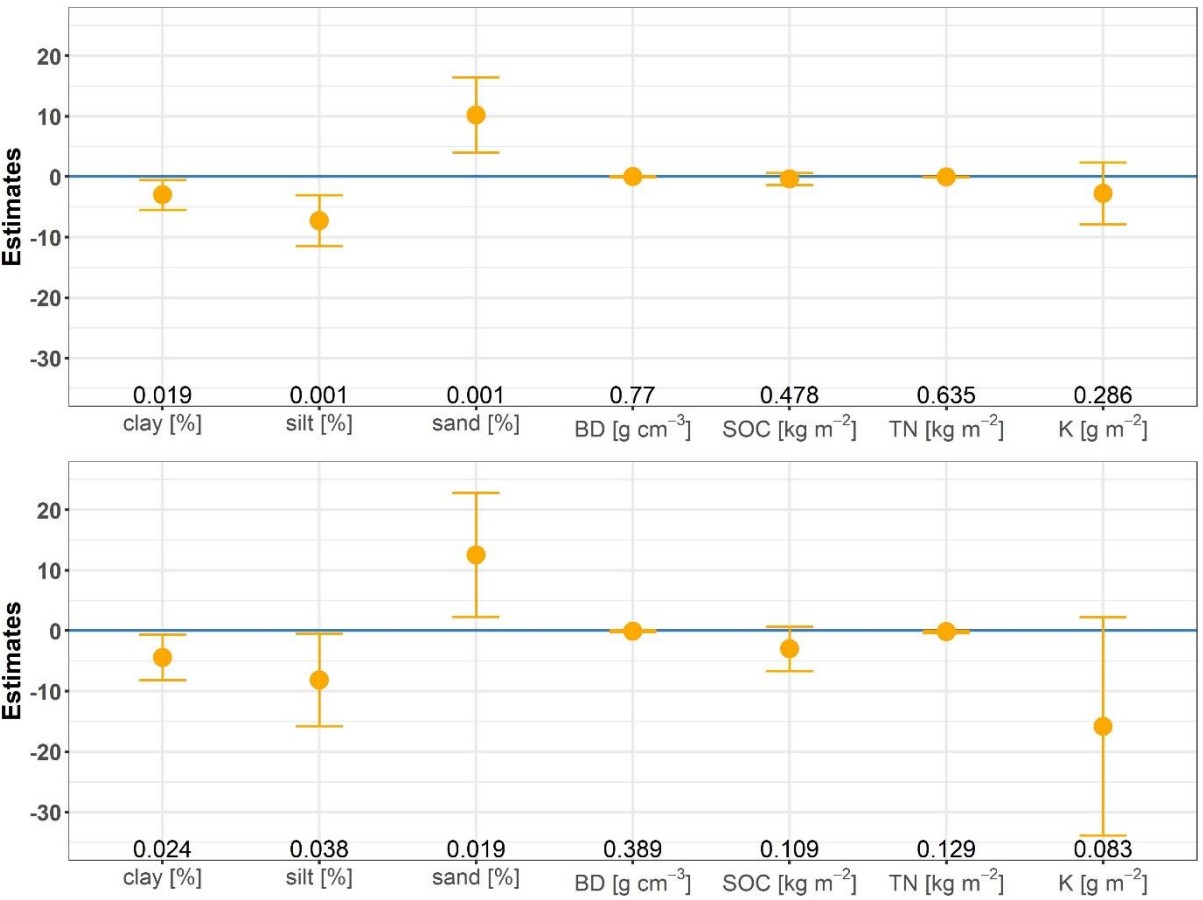





**(n = 305). P values indicated below the plots were calculated for the differences between the estimates for soils under DNOR and DWIR. Data for exchangeable Mg were left out for better readability (estimate: +4.5; confidence intervals: +21.2/-12.2; p = 0.596). Bottom: Estimates and 95 % confidence intervals of linear regression models, comparing the uppermost 10 cm of the soil profiles under DNOR (orange) to those under DWIR (blue) (n = 54). P values indicated below the plots were calculated for the differences between the estimates for soils under DNOR and DWIR. Data for exchangeable Mg were left out for better readability (estimate: -**

**32.2; confidence intervals: +16.5/-80.9; p = 0.184). BD = bulk density, SOC = soil organic carbon, TN = total nitrogen, K = exchangeable potassium.**

### 3.3 Hydrological soil properties

In general, all soils had high hydraulic conductivities, due to their predominantly sandy texture. A site-wise comparison between soils under DWIR and DNOR showed that soils under DWIR had higher hydraulic conductivities in their uppermost

310 horizons (Fig. 8 bottom). For example, the hydraulic conductivities of soil profiles 31 and 32 (DNOR) were below $1.5 \cdot 10^{-3}$ cm $s^{-1}$ whereas those of profiles 33 – 36 (DWIR) ranged between 1.5 and $3 \cdot 10^{-3}$ cm $s^{-1}$. Profiles 38 and 40 (DNOR) showed hydraulic conductivities between $2.5 \cdot 10^{-3}$ cm $s^{-1}$ and $3.5 \cdot 10^{-3}$ cm $s^{-1}$ in their uppermost horizons, while the hydraulic conductivities of profiles 37 and 41 (DWIR) exceeded $5 \cdot 10^{-3}$ cm $s^{-1}$. Soils under FOR showed similar hydraulic conductivities, with a maximum of $3.5 \cdot 10^{-3}$ cm $s^{-1}$ (Fig. 8 top). Soils under FOR had lower plant-available field capacities than soils under

315 DWIR and DNOR (Fig. 8). Except for one subsoil horizon, plant-available field capacity ranged from 12 vol. % to 22 vol. % in soils under FOR, and from 12 vol. % to 33 vol. % in soils under DWIR and DNOR. The differences between soils under DWIR and DNOR became clearer in a site-wise comparison. For example, plant-available field capacities of profiles 39 and 40 (DNOR) ranged between 12 vol. % and 17 vol. %, whereas profile 37 (DWIR) had plant-available field capacities between 17 vol. % and 26 vol. %. However, the limited number of samples analysed for plant-available field capacity does not allow

320 for a final statement on differences between the sites with respect to this factor.





**Figure 8: Hydraulic conductivities (a) and plant-available field capacities (b) of soils under near-natural forest (FOR), and hydraulic conductivities (c) and plant-available field capacities (d) of soils under disturbed forest with regrowth of trees (DWIR; left) and under disturbed forest showing no regrowth of trees (DNOR; right). Soil horizons in light grey colour were not subjected to these measurements.**

Water repellency of topsoils under different vegetation categories did not show any differences (Fig. A2, appendix). Topsoil samples from all vegetation categories included samples, on which the water drop only persisted for one or a few seconds, and samples on which the drop persisted for more than 1000 seconds.

## 4 Discussion

### 4.1 Chemical soil properties

In general, Siberian larch has low requirements with respect to chemical soil properties (Dylis, 1947). It can grow on a wide range of soils and may colonise even rather fresh sediments (Timoshok et al., 2014). Siberian larch can also adapt to low nutrient availability by enhanced soil nutrient exploitation (Lebedev, 2012; Chernyshenko and Vasilyev, 2019). Most importantly, it needs nitrogen (Lebedev and Lebedev, 2010; Chernyshenko and Vasilyev, 2019), like many other larch species including *Larix gmelinii* (Schulze et al., 1995; Liang et al., 2014), *Larix kaempferi* (Leyton, 1956; Kayama et al., 2009) and hybrids such as *Larix x eurolepis* (Pâques, 1994). Also a lack in K and Mg may limit the growth of larch species on poor, sandy soils (Leyton, 1956; Phu, 1975), whereas growth limitation by insufficient P supply has not yet been reported. Generally, differences in the vitality and growth of larch are more commonly caused by climatic or hydrological differences than by nutrient limitations (Fiedler et al., 1980; Stüber, 1998; Viers et al., 2013).

Because of the low nutrient requirements of Siberian larch and since we did not detect any significant differences in the chemical properties of soils under d̲isturbed forest w̲it̲h r̲egrowth of trees (DWIR) and under d̲isturbed forest showing n̲o r̲egrowth of trees (DNOR), we conclude that chemical soil properties are not responsible for the differences in post-disturbance regrowth of Siberian larch in our study area. The uppermost 10 cm of the soils under FOR had even significantly smaller N stocks and lower contents of exchangeable Ca, Mg and K than the uppermost 10 cm of the soils under DNOR. Nevertheless, saplings and young trees were growing in the FOR areas, which confirms that the nutrient supply of all analysed soils fulfilled the needs of Siberian larch. Furthermore, the nutrient stocks were similar or even higher compared to those reported from other Siberian larch forests and from forests of various other larch species (Fiedler et al., 1980; Hwang and Son, 2006; Kayama et al., 2009; Park et al., 2009; Lebedev and Lebedev, 2010; Watanabe et al., 2012; Wang et al., 2014).

### 4.2 Physical soil properties and soil hydrology

Soils under DWIR had significantly more silt and clay, and thus higher plant-available field capacity (Amelung et al., 2018) than soils under DNOR, which were considerably sandier. As mean annual precipitation in Tosontsengel is only 200-250 mm, lack of water represents a major limitation for tree growth (Dulamsuren et al., 2009; 2010; 2011; Chenlemuge et al., 2015b;





2015a). Under these climatic conditions, it is plausible that soil texture and corresponding plant-available field capacity are

dominant factors controlling the post-disturbance tree regrowth pattern. The measured plant-available field capacities confirmed the difference between the soils under DWIR and DNOR. Moreover, the data of the entire soil profiles showed that the texture of soils under DWIR was similar to that of soils under near-natural forest (FOR), and that the texture of soils under DNOR was similar to that of soils under steppe (STE). Also, the PCA confirmed the similarities between soils under DNOR and STE, thus pointing to a potential risk of DNOR to shift from forest to steppe vegetation. Such potential shift has already

been predicted for the forest-steppe in the Khentey Mountains, Mongolia (Dulamsuren and Hauck, 2008; Dulamsuren et al., 2009).

Hydraulic conductivity was higher in the uppermost horizons of soils under DWIR compared to those under DNOR. Trees may benefit from this difference for two reasons. Firstly, rapid infiltration through the uppermost horizons reduces evaporation loss. Secondly, grasses and herbs have a dense but shallow root system and compete with tree roots for water at shallow soil

depth (Müller-Hohenstein, 1981; Breckle et al., 1994). As tree roots reach deeper down the soil, they benefit from rapid water infiltration below the depth of the roots of grasses and herbs. Lange et al. (2015) carried out irrigation experiments on grass-dominated south-facing slopes and forest-dominated north-facing slopes in the Mongolian forest-steppe. On the south-facing slopes, most of the water was either taken up by grass or evaporated, and the remaining water reached only 5 cm soil depth. On the north-facing slopes, the water percolated down to the permafrost table. Although the contrasting exposition also affected

this experiment, these results confirm the competitiveness of grasses for water in the upper soil horizons and the relevance of soil hydraulic conductivity for tree growth in water-limited environments.

We did not detect a positive effect of permafrost on post-disturbance tree regrowth, as permafrost was neither encountered under DWIR nor under DNOR. This is in agreement with observations by Kopp et al. (2014), who reported absence of permafrost five years after severe forest fire. The same authors moreover measured increased soil moisture in soils of burned

sites, compared to soils under forest, which they attributed to the absence of tree transpiration. In contrast, Park et al. (2009) stated a decrease in soil moisture after fire and logging, arguing that evaporation loss from the bare soils and deterioration of physical soil properties exceed the decrease in transpiration. In our study area, we also observed rather decreased soil moisture at sites disturbed by fire and logging, compared to near-natural forest sites. An additional decline in soil moisture after fire can be induced by water repellency that may increase surface run off (DeBano, 2000; Mataix-Solera and Doerr, 2004). However,

we did not observe a significant difference in water repellency between the four vegetation categories, possibly because of the high frequency of fires in in our study area, and associated enhanced erosion (Goldammer, 2002; Hessl et al., 2012). As water repellency occurred irregularly across the whole study area, including DWIR and DNOR, it seemed to be not relevant for the post-disturbance tree recovery pattern in our study area.

### 4.3 Further relevant environmental factors

Other factors besides soil hydrology that influence the post-disturbance tree regrowth pattern, include in particular relief and human activity. Relief-induced water gains in concave positions may support tree growth even where soil properties seem to





be unfavourable. Similarly, water losses through divergence in convex positions may hamper tree growth, even where soil properties seem to be suitable. Also, small tree patches that survived a disturbance, can have a positive influence on post-disturbance tree regrowth. In our study area, we observed enhanced fructification of trees that had survived forest fires,
initiating tree regrowth in belts around small tree patches that had persisted. We explain this observation by the shading of the remaining adult trees that reduces evaporation and creates a more even microclimate on the ground, and by increased availability of intact seeds in the direct surrounding of mature trees (Dugarjav, 2006).

In contrast, human activity, especially logging and pastoral pressure, may inhibit post-disturbance forest recovery (Khishigjargal et al., 2013; Dulamsuren et al., 2014; Khansaritoreh et al., 2017b). In particular, goats grazing in areas of burned
forest or in forest stands that have previously been opened by partial logging, hamper the growth of seedlings and damage young trees (Sankey et al., 2006). Thus, human impact and soil hydrology are the two key factors controlling the post-disturbance tree recovery pattern (Fig. 9).

Loss of the shading by the forest canopy, of the forest microclimate and the insulating organic layer in turn lead to permafrost degradation. Also, the presently ongoing climate change accelerates permafrost decline in Mongolia. Thus, this essential soil
water reservoir is likely to disappear (Sharkhuu and Sharkhuu, 2012). As the meltwater from the active layer above the permafrost table can support tree growth even at sites where soil water storage is below the threshold for tree growth, permafrost decline may also contribute to the decrease in forest area in the Mongolian forest-steppe. In some of our investigated sites, this has already happened.

Klinge et al. (2020) estimated that the potential forest distribution in the Khangai Mountains is three times larger than the
actual forest area. This discrepancy indicates that forest fire and human activity already caused a considerable decline of forested areas.





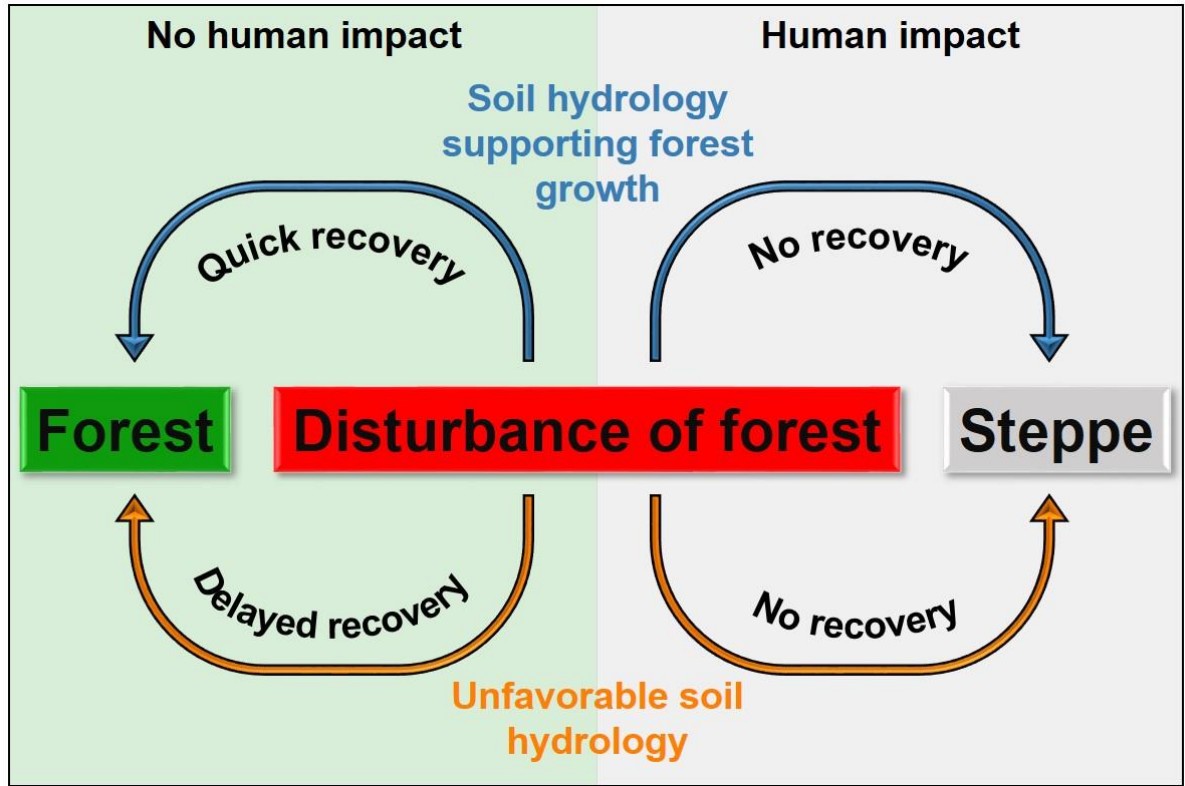

**Figure 9: Possible development pathways of disturbed forest areas in the Mongolian forest-steppe, with respect to hydrological soil properties and further environmental factors.**

## 5 Conclusions

This study showed that the post-disturbance regrowth pattern of trees in the central Mongolian forest-steppe, a highly sensitive, semi-arid ecotone, is largely controlled by soil hydrology. With respect to our hypotheses established in the beginning, we conclude that:

1) Differences in post-disturbance tree regrowth depend mainly on soil texture. Loamy soils have high plant-available field capacity and are thus able to store sufficient amounts of water for post-disturbance tree regrowth. In contrast, sandy soils cannot store enough water for recovery of tree vegetation under the present environmental conditions.

2) Post-disturbance tree regrowth is favoured by higher hydraulic conductivity in the uppermost soil horizons, as rapid percolation through the upper soil horizons reduces evaporation loss. In addition, trees benefit from the reduced competition for water with grasses and herbs that results from rapid water percolation into the subsoil.

3) Nutrient limitation is not relevant for the uneven post-disturbance tree regrowth pattern in our study area, as the dominant tree species, Siberian larch, has rather low nutrient requirements.





4) Meltwater from the active layer above the permafrost table provides additional water for near-natural forests. However, it cannot be a relevant factor for the uneven post-disturbance regrowth pattern of Siberian larch in our study area, because permafrost was encountered neither under disturbed forests with nor without tree regrowth.

Under the given climatic conditions, water limitation is the predominant factor controlling tree growth in the semi-arid Mongolian forest-steppe. Therefore, even small differences in soil properties that lead to an increase in the amount of plant-available water, can be decisive for post-disturbance tree regrowth.

Disturbances by severe fire or clear-cutting result in loss of permafrost and thus loss of an additional water reservoir during summer. Such additional water source can be critical at sites, where soil water storage alone is slightly below the threshold for 430 tree growth. Therefore, such sites are particularly prone to an irreversible shift from forest to steppe after strong disturbance, involving loss of permafrost. Moreover, we expect that many sites where soil hydrology allows for post-disturbance tree regrowth under the present climatic conditions, will lose their ability for post-disturbance forest recovery in the course of the ongoing climate change.

Climate projections predict that the decrease of permafrost will continue, and drought events will become both more frequent 435 and more intensive. At the same time, it seems unlikely that the pressure on forests in the Mongolian forest-steppe by fire, logging and grazing will decrease, unless explicit and effective measures are taken. Consequences of this unfavourable interplay will most likely include a decrease in the growth rates of trees, and a further considerable and irreversible decline of forested area. Therefore, regulation of human impact on the forests in the Mongolian forest-steppe is mandatory, if a further irreversible loss of forest area shall be avoided.

**6 Appendix A**

**Table A1: General information on the soil profiles. Abbreviations are according to the Guidelines for Soil Description (FAO, 2006). \*: If permafrost is present, the depth of the permafrost table equals the profile depth.**





| Vegetation category | Profile ID | Coordinates N | Coordinates E | Elevation [m a.s.l.] | Inclination [%] | Exposition [°] | Bedrock | Main sediment type | Pendants on rock fragments | Profile depth [cm] | Presence of permafrost* | Disturbance by fire | Disturbance by logging | Soil type [based on WRB] |
|---|---|---|---|---|---|---|---|---|---|---|---|---|---|---|
| FOR | 1 | 48° 38.681' | 98° 22.141' | 1915 | 24.5 | 320 | granite | sand sheet | no | 165 | yes | no | no | Eutric **Cambisol** (Katoarenic, Bathygelic, Epiloamic, Nechic, Ochric, Endoraptic) |
| FOR | 2 | 48° 38.700' | 98° 22.034' | 1907 | 26.5 | 340 | granite | sand sheet | no | 140 | yes | no | no | Eutric **Cambisol** (Katoarenic, Bathygelic, Amphiloamic, Nechic, Ochric) |
| STE | 3 | 48° 38.929' | 98° 22.122' | 1882 | 28.5 | 301 | granite | bedrock | no | 90 | no | no | no | Skeletic Cambic Endoleptic **Phaeozem** (Amphiarenic, Epiloamic, Nechric) |
| STE | 4 | 48° 38.377' | 98° 22.202' | 1917 | 25.75 | 292 | granite | colluvium | no | 90 | no | no | no | Endoskeletic **Phaeozem** (Amphiarenic, Colluvic, Epiloamic, Nechric, Epiraptic) |
| STE | 5 | 48° 38.736' | 98° 21.853' | 1851 | 20 | 307 | granite | colluvium | no | 110 | no | no | no | Endoskeletic **Chernozem** (Katoarenic, Colluvic, Amphiloamic, Amphiraptic) |
| STE | 6 | 48° 38.216' | 98° 22.408' | 1979 | 24.5 | 300 | granite/gneiss | slope deposit | no | 120 | no | no | no | Endoskeletic Endoleptic **Phaeozem** (Katoarenic, Colluvic, Epiloamic, Nechric, Epiraptic) |
| FOR | 7 | 48° 38.181' | 98° 22.487' | 2007 | 31 | 304 | granite/gneiss | sand sheet | no | 105 | no | no | low | Eutric Endoskeletic **Cambisol** (Katoarenic, Epiloamic, Nechic, Ochric, Epiraptic) |
| FOR | 8 | 48° 38.206' | 98° 22.511' | 2022 | 32 | 305 | granite/gneiss | sand sheet | no | 120 | no | no | low | Eutric Endoskeletic **Cambisol** (Katoarenic, Amphiloamic, Nechic, Ochric, Endoraptic) |
| FOR | 9 | 48° 38.770' | 98° 22.074' | 1868 | 16 | 301 | granite | sand sheet | no | 180 | yes | no | moderate | Cambic **Phaeozem** (Katoarenic, Colluvic, Katoloamic, Nechic, Endoraptic, Bathyprotocalcic, Bathygleyic, Bathyturbic) |
| FOR | 10 | 48° 38.748' | 98° 22.169' | 1900 | 26.5 | 303 | granite | slope deposit | no | 100 | no | no | moderate | Haplic **Phaeozem** (Endoarenic, Colluvic, Amphiloamic, Nechric, Amphiraptic) |
| FOR | 11 | 48° 37.766' | 98° 21.215' | 1938 | 23.5 | 335 | granite/gneiss | slope deposit | manganese | 60 | yes | no | no | Endoskeletic Cambic Folic Reductaquic **Cryosol** (Epiarenic, Eutric, Amphiloamic, Nechic, Ochric, Epiraptic) |
| FOR | 12 | 48° 37.775' | 98° 21.277' | 1943 | 28.5 | 317 | granite/gneiss | slope deposit | no | 80 | yes | no | no | Endoskeletic Folic Reductaquic **Cryosol** (Amphiarenic, Eutric, Epiloamic, Nechic, Ochric, Amphiraptic) |
| FOR | 13 | 48° 37.258' | 98° 21.517' | 1974 | 27.5 | 335 | granite/gneiss | slope deposit | manganese | 60 | yes | no | no | Endoskeletic Folic Reductaquic **Cryosol** ( Eutric, Loamic, Nechic, Ochric, Epiraptic) |
| FOR | 14 | 48° 37.322' | 98° 21.624' | 1969 | 27 | 326 | granite/gneiss | slope deposit | no | 80 | yes | no | no | Endoskeletic Folic **Cryosol** (Epiarenic, Eutric, Amphiloamic, Nechic, Ochric, Epiraptic) |
| FOR | 15 | 48° 37.728' | 98° 21.233' | 1947 | 25 | 324 | granite/gneiss | slope deposit | manganese | 94 | yes | no | no | Endoskeletic Cambic Folic Reductaquic **Cryosol** (Amphiarenic, Eutric, Amphiloamic, Nechic, Ochric, Epiraptic) |
| FOR | 16 | 48° 37.787' | 98° 21.210' | 1913 | 23.5 | 357 | granite/gneiss | slope deposit | manganese | 66 | yes | no | low | Cambic Folic Reductaquic **Cryosol** (Amphiarenic, Eutric, Amphiloamic, Nechic, Ochric, Epiraptic) |
| FOR | 17 | 48° 37.816' | 98° 21.208' | 1908 | 22 | 359 | granite/gneiss | slope deposit | no | 80 | yes | no | low | Endoskeletic Cambic Reductaquic **Cryosol** (Endoarenic, Eutric, Epiloamic, Nechic, Ochric, Epiraptic) |
| FOR | 18 | 48° 37.755' | 98° 21.280' | 1951 | 34.5 | 351 | granite/gneiss | slope deposit | manganese | 100 | yes | no | no | Endoskeletic Cambic Reductaquic **Cryosol** (Epiarenic, Eutric, Amphiloamic, Nechic, Ochric, Epiraptic) |
| FOR | 19 | 48° 37.799' | 98° 21.268' | 1930 | 30 | 358 | granite/gneiss | slope deposit | manganese | 80 | yes | no | no | Endoskeletic Folic Reductaquic **Cryosol** (Eutric, Loamic, Nechic, Ochric, Epiraptic) |
| FOR | 20 | 48° 37.825' | 98° 21.271' | 1914 | 24 | 1 | granite/gneiss | slope deposit | no | 65 | yes | no | no | Endoskeletic Cambic Folic Reductaquic **Cryosol** (Endoarenic, Eutric, Anoloamic, Nechic, Ochric, Endoraptic) |
| FOR | 21 | 48° 37.700' | 98° 21.276' | 1990 | 39 | 315 | granite/gneiss | slope deposit | no | 90 | yes | no | no | Endoskeletic Reductaquic **Cryosol** (Amphiarenic, Eutric, Amphiloamic, Nechic, Ochric, Epiraptic) |
| FOR | 22 | 48° 37.830' | 98° 21.208' | 1900 | 11 | 334 | granite/gneiss | slope deposit | no | 90 | yes | no | low | Endoskeletic Folic Reductaquic **Cryosol** (Endoarenic, Eutric, Amphiloamic, Nechic, Ochric, Amphiraptic) |
| FOR | 23 | 48° 37.736' | 98° 21.290' | 1977 | 37 | 345 | granite/gneiss | slope deposit | no | 100 | yes | no | no | Endoskeletic Reductaquic **Cryosol** (Amphiarenic, Eutric, Epiloamic, Nechic, Ochric, Epiraptic) |
| FOR | 24 | 48° 37.843' | 98° 21.256' | 1908 | 12 | 352 | granite/gneiss | slope deposit | no | 90 | yes | no | low | Endoskeletic Folic Reductaquic **Cryosol** (Endoarenic, Eutric, Amphiloamic, Nechic, Ochric, Epiraptic) |
| FOR | 25 | 48° 37.649' | 98° 21.231' | 1994 | 28.5 | 336 | granite/gneiss | slope deposit | no | 70 | no | no | no | Eutric Endoskeletic **Cambisol** (Gelic, Gelistagnic, Loamic, Nechric, Ochric, Epiraptic) |
| FOR | 26 | 48° 37.669' | 98° 21.199' | 1982 | 29 | 346 | granite/gneiss | slope deposit | no | 70 | yes | no | no | Endoskeletic Folic Reductaquic **Cryosol** (Epiarenic, Eutric, Amphiloamic, Nechic, Ochric, Epiraptic) |
| FOR | 27 | 48° 37.697' | 98° 21.184' | 1977 | 29 | 356 | granite/gneiss | slope deposit | no | 58 | yes | no | no | Cambic Folic **Cryosol** (Eutric, Loamic, Nechic, Ochric, Epiraptic) |
| FOR | 28 | 48° 37.725' | 98° 21.161' | 1957 | 24.5 | 352 | granite/gneiss | slope deposit | no | 80 | yes | no | no | Endoskeletic Folic Reductaquic **Cryosol** (Amphiarenic, Eutric, Amphiloamic, Nechic, Ochric, Epiraptic) |
| FOR | 29 | 48° 37.768' | 98° 21.135' | 1943 | 25.5 | 353 | granite/gneiss | slope deposit | no | 80 | yes | no | no | Endoskeletic Folic Reductaquic **Cryosol** (Epiarenic, Eutric, Amphiloamic, Nechic, Ochric, Epiraptic) |
| FOR | 30 | 48° 37.787' | 98° 21.127' | 1915 | 15 | 351 | granite/gneiss | slope deposit | no | 70 | yes | no | low | Endoskeletic Cambic Folic Reductaquic **Cryosol** (Eutric, Loamic, Nechric, Ochric, Epiraptic) |
| DNOR | 31 | 48° 27.636' | 98° 13.686' | 2046 | 22 | 316 | granite/gneiss | slope deposit | calcium carbonate | 100 | no | severe | moderate | Eutric Endoskeletic **Cambisol** (Endoarenic, Amphiloamic, Nechric, Ochric, Amphiraptic) |
| DNOR | 32 | 48° 27.640' | 98° 13.730' | 2055 | 25.5 | 344 | granite/gneiss | slope deposit | calcium carbonate | 80 | no | severe | moderate | Eutric Endoskeletic **Cambisol** (Endoarenic, Colluvic, Epiloamic, Nechric, Ochric, Amphiraptic) |
| DWIR | 33 | 48° 27.549' | 98° 13.576' | 2055 | 25 | 312 | granite/gneiss | slope deposit | calcium carbonate | 80 | no | severe | moderate | Eutric Endoskeletic **Cambisol** (Colluvic, Loamic, Nechric, Ochric, Epiraptic) |
| DWIR | 34 | 48° 27.595' | 98° 13.723' | 2079 | 32.5 | 332 | granite/gneiss | slope deposit | manganese, calcium carbonate | 90 | no | severe | moderate | Eutric Endoskeletic **Cambisol** (Protocalcic, Colluvic, Loamic, Nechric, Amphiraptic) |
| DWIR | 35 | 48° 27.597' | 98° 10.886' | 2077 | 26.5 | 322 | granite/gneiss | slope deposit | no | 90 | no | severe | moderate | Eutric Endoskeletic **Cambisol** (Loamic, Nechric, Ochric, Epiraptic) |
| DWIR | 36 | 48° 27.609' | 98° 10.914' | 2069 | 29.5 | 320 | granite/gneiss | slope deposit | no | 100 | no | severe | moderate | Eutric Endoskeletic **Cambisol** (Loamic, Nechric, Ochric, Epiraptic) |
| DWIR | 37 | 48° 28.784' | 98° 19.246' | 2088 | 35 | 324 | granite/gneiss | slope deposit | no | 110 | no | severe | moderate | Eutric Endoskeletic **Cambisol** (Amphiarenic, Colluvic, Epiloamic, Nechric, Ochric, Epiraptic) |
| DNOR | 38 | 48° 28.790' | 98° 19.074' | 2004 | 41 | 328 | granite/gneiss | colluvium | no | 100 | no | severe | moderate | Endoskeletic **Phaeozem** (Amphiarenic, Colluvic, Epiloamic, Nechric, Ochric, Amphiraptic) |
| DNOR | 39 | 48° 28.732' | 98° 18.980' | 2018 | 36 | 328 | granite/gneiss | slope deposit | calcium carbonate | 115 | no | severe | severe | Eutric Endoskeletic **Cambisol** (Amphiarenic, Colluvic, Epiloamic, Nechric, Ochric, Amphiraptic) |
| DNOR | 40 | 48° 28.335' | 98° 19.140' | 2040 | 36.5 | 320 | granite/gneiss | slope deposit | manganese, calcium carbonate | 100 | no | severe | moderate | Eutric Endoskeletic **Cambisol** (Amphiarenic, Colluvic, Epiloamic, Nechric, Ochric, Amphiraptic) |
| DWIR | 41 | 48° 28.345' | 98° 19.228' | 2053 | 42.5 | 330 | granite/gneiss | slope deposit | calcium carbonate | 110 | no | severe | moderate | Eutric Endoskeletic **Cambisol** (Amphiarenic, Colluvic, Epiloamic, Nechric, Ochric, Amphiraptic) |
| DNOR | 42 | 48° 33.181' | 98° 13.362' | 1893 | 23 | 330 | granite | slope deposit | no | 110 | no | no | severe | Haplic **Kastanozem** (Amphiarenic, Cambic, Colluvic, Epiloamic, Nechric, Epiraptic) |
| DNOR | 43 | 48° 24.506' | 98° 18.295' | 1912 | 35 | 17 | granite | slope deposit | no | 125 | no | no | severe | Skeletic **Phaeozem** (Katoarenic, Colluvic, Amphiloamic, Nechric, Amphiraptic) |
| DWIR | 44 | 48° 48.525' | 98° 19.850' | 2093 | 25.5 | 320 | granite/gneiss | slope deposit | no | 100 | no | severe | severe | Eutric Endoskeletic **Cambisol** (Amphiarenic, Epiloamic, Nechric, Ochric, Amphiraptic) |
| DNOR | 45 | 48° 48.371' | 98° 19.351' | 2059 | 23 | 324 | granite/gneiss | slope deposit | no | 90 | no | severe | severe | Eutric Endoskeletic **Cambisol** (Amphiarenic, Epiloamic, Nechric, Ochric, Epiraptic) |
| DNOR | 46 | 48° 48.400' | 98° 19.136' | 2002 | 28 | 330 | granite/gneiss | slope deposit | manganese | 105 | no | severe | severe | Eutric Endoskeletic **Cambisol** (Katoarenic, Epiloamic, Nechric, Ochric, Amphiraptic) |
| DWIR | 47 | 48° 52.582' | 98° 19.482' | 2006 | 32.5 | 315 | granite/gneiss | bedrock | manganese | 80 | no | moderate | moderate | Eutric Endoskeletic Endoleptic **Cambisol** (Amphiarenic, Amphiloamic, Nechric, Ochric, Epiraptic) |
| STE | 48 | 48° 16.462' | 98° 21.224' | 2045 | 3 | 305 | granite | slope deposit | calcium carbonate | 130 | no | no | no | Haplic **Kastanozem** (Cambic, Loamic, Amphiraptic) |
| DWIR | 49 | 48° 16.438' | 98° 20.856' | 2043 | 26.5 | 325 | granite | slope deposit | manganese | 90 | no | severe | moderate | Eutric Endoskeletic **Cambisol** (Amphiarenic, Epiloamic, Nechric, Ochric, Epiraptic) |
| DWIR | 50 | 48° 22.427' | 98° 38.329' | 2039 | 23 | 357 | granite | slope deposit | no | 105 | no | severe | moderate | Endoskeletic **Phaeozem** (Katoarenic, Colluvic, Amphiloamic, Nechric, Amphiraptic) |
| DNOR | 51 | 48° 22.293' | 98° 38.313' | 2098 | 34 | 4 | granite | slope deposit | no | 110 | no | severe | moderate | Endoskeletic **Phaeozem** (Katoarenic, Colluvic, Amphiloamic, Nechric, Amphiraptic) |
| DWIR | 52 | 48° 22.113' | 98° 38.247' | 2213 | 30.5 | 355 | granite | bedrock | no | 52 | no | severe | no | Endoskeletic Endoleptic **Phaeozem** (Loamic, Nechric, Epiraptic) |
| DWIR | 53 | 48° 41.506' | 98° 27.930' | 1934 | 45 | 290 | gneiss | slope deposit | manganese | 95 | no | moderate | moderate | Skeletic Endoleptic **Phaeozem** (Amphiarenic, Epiloamic, Nechric, Epiraptic) |
| DWIR | 54 | 48° 41.471' | 98° 28.073' | 2033 | 44.5 | 291 | gneiss | slope deposit | manganese | 75 | no | moderate | moderate | Endoskeletic Endoleptic **Phaeozem** (Amphiarenic, Epiloamic, Nechric, Epiraptic) |





**Figure A2: Violin plots and boxplots for water repellency (expressed as water drop infiltration time in seconds; n = 53), elevation (n = 54), inclination (n = 54), and exposition (n = 54) of the soils under the four vegetation categories near-natural forest (FOR, green), steppe close to the forest (STE, grey), disturbed forest with regrowth of trees (DWIR, blue) and disturbed forest showing no regrowth of trees (DNOR, orange). P values indicated above the plots were calculated for differences between the arithmetic means (NS.: p = 1; ns: p > 0.05; \*: p ≤ 0.05; \*\*: p ≤ 0.01; \*\*\*: p ≤ 0.001; \*\*\*\*: p ≤ 0.0001).**

**Code/Data availability:**

See supplementary material (data.xlsx).



**Authors contribution:**

Michael Klinge and Daniela Sauer designed the research project. All authors carried out the field work together. Florian
Schneider, Tino Peplau and Jannik Brodthuhn carried out the laboratory work. Florian Schneider prepared the manuscript with
contributions from all co-authors.

**Competing interests:**

The authors declare that they have no conflict of interest.

**Acknowledgements**

We thank Dr. Choimaa Dulamsuren and Prof. Uudus Bayarsaikhan for their invaluable help in organising and conducting our
fieldwork, and Ms. Daramragchaa Tuya for her great support of our research. We are also grateful to our Mongolian colleagues
Mr. Amarbayasgalan, Mr. Enkhjargal, Mr. Enkh-Agar, and Ms. Munkhtuya; we appreciated their hospitality and help with the
fieldwork. Our thanks also go to the German students Janin Klaassen, Kim Lena Arndt and Tim Rollwage for their great
commitment during the fieldwork in Mongolia. Furthermore, we want to thank Dr. Jürgen Grotheer, Petra Voigt, Anja Södje
and Eike Sebode for their excellent work in the laboratory. We also thank Lukáš Banyi for the essential support in using the R
project for statistical computing.

The project was funded by the Deutsche Forschungsgemeinschaft (DFG), project number 385460422.

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
