# Peer review of "Hydrological soil properties control tree regrowth after forest disturbance in the forest-steppe of central Mongolia"

_SOIL, 2020_

## Referee Comment (RC1) · Anonymous Referee #1 · 24 Nov 2020

The authors of this study aimed to investigate the effect of various soil properties on tree regrowth in disturbed Mongolian forest steppe. The study is well done, written in good language, and sections are well arranged. The study will be of considerable interest for a broad international community. Most of my comments, which are listed below, are of technical nature. I therefore recommend minor revision.

GENERAL COMMENTS - Good language

SPECIFIC COMMENTS - Key words seem a bit unbalanced after reading the Abstract; e.g. permafrost was said to have no major effect; moreover there is nothing about hydraulic features in the key words although this is the main aspect of the manuscript.

- Discussion: Figure 9 seems a nice idea, but could show more information. E.g., maybe it would make sense to hint at the effect of certain factors by making the arrows in different width? Moreover, the favorable or unfavorable soil hydrological conditions might be shown on the left side as an axis. These are just suggestions. - The authors differed for each site between low, moderate and severe disturbance. However, this was not included comprehensively in the Discussion section.

TECHNICAL CORRECTIONS - Introduction and generally: please do not cite more than 3 studies for one certain context or statement. - Material and methods: please give not only the name of the used analytical instruments but also company. - Chapter 4.2 (and generally in the Discussion section): I think it would be good to refer to the respective figures again, instead of just referring to 'the PCA'. - The following comments refer to the References: > 90 references is too much. I think the list could be shortened by ca. 25%, see also my first comment for the Introduction - In my opinion, it is not appropriate to cite educational books such as Scheffer/Schachtschabel, 'Bodenkundliches Praktikum', 'Spezielle Ökologie der Gemäßigten und Arktischen Zonen Euro-Nordasiens', 'Skript Bodenphysikalische Versuche', 'Die Landschaftsgürtel der Erde', even more so as they are written in German language. The same is true for PhD theses (Sommer 2000, Stüber 1998). - Why are some papers in the list written with DOI, others not? - Also, I think it is not nice to cite 'Deutsches Institut für Normung' for the applied methods, especially in a manuscript which will be of interest for an international community. - The 'Guidelines for soil description' are cited correctly in the text, but written incorrectly in the references list. - For the citation Klinge et al. 2020 the journal is missing. - The authors did not use uniform notation for the journals (abbreviated or not, abbreviations with or without '.'). - Generally, I'm not so happy about the large portion of literature in non-English language (German, Russian, Mongolian; together almost 25% of references!). - Figures: 9 figures might be a bit too much. Maybe some of them could be moved to the Supplement, e.g. the climate diagram, or could be converted into a table?

---

## Referee Comment (RC2) · Anonymous Referee #2 · 1 Dec 2020

General comments

The authors assessed the relative importance of different soil properties on the forest regrowth success after disturbance in central Mongolia. The main problematic is the sustainability of wood resource in a semi-arid region, which, among other factors, depends on soil properties. This study is at the crossroad of soil science/climate change/human activities. As such, the questions addressed in this paper fall within the scope of SOIL.

The hypotheses are clearly stated and the authors neatly wrap everything up by evaluating the hypotheses in the conclusion.

[Figure]

The methods are clearly described and can be reproduced. The methods and the statistical analyses used are adequate to test the hypotheses stated. The tools are not necessarily novel (basic soil analyses). The novelty resides in the studied ecosystems: forests and steppes in Mongolia. This is shown by a rapid bibliometric analysis (on the 1st of December) in a popular scientific search engine: "European forest" - 3410000 results, "German forest" - 2320000 results, "Mongolian forest" - 60900 results. The resulting database is thus original and valuable for the soil science in general as it gives basic pedologic information on a relatively less studied area of the world.

The soils are well described and the descriptions follow FAO guideline. The results are clearly represented, other than the figure 8 (see specific comment). Most of the discussion is relevant and supported by the results. However, key questions arise about the figure 8, which weaken the subsequent parts in the discussion based on the results presented in this figure.

Specific comment

My main concerns is about the figure 8, representing hydraulic conductivity and field capacity results. Important parts of the discussion and the conclusions are based on these results. However, the way these results are presented now do not clearly support the discussion/conclusion.

1) This is a site wise comparison without any statistical analysis, which makes difficult to draw any conclusion.

2) In Fig. 8c, the number of DWIR site is superior to DNOR sites: 10 and 6 sites for DWIR and DNOR, respectively. Then, in Fig. 8d, there are 5 sites for each situation. Is there any explanation of the different number of samples, within one analysis (hydraulic conductivity) and between analyses (more than 5 for hydraulic conductivity, 5 for field capacity)? How the sites for these analyses were chosen? Can the author assure that choice of the sites did not generate any bias? The authors should give information to clarify these points.

3) In the way data are presented in Fig. 8c and 8d, it is not easy to analyse the difference between DWIR and DNOR. What you see in a first sight is just orange and red boxes on the left hand side of the dotted blue line and orange and red boxes on the right hand side of the dotted blue line... Then, you have to go back to the map in Fig. 1 to check which DWIR and DNOR sites are closed by so you can start making groups to ease the comparison, like 37+41 for DWIR to compare to 38+40, for hydraulic conductivity. When you compare these 2 groups, you can see that hydraulic conductivity tend to be higher in DWIR compared to DNOR. When you compare field capacity of 37 (DWIR) to 39+40 (DNOR), you can see that indeed field capacity tend to be higher in DWIR situation.

The authors should try to make statistical analysis and improve the presentation of the data (e.g. by group of sites as suggested above) to have results that better supports the conclusion reached.

Technical comment

The word "ecotone" is used several times. Following the Oxford Dictionary of Ecology, ecotone is "a narrow and fairly sharply defined transition zone between two or more different communities. Such edge communities". It seems there is a misuse of the word ecotone in this article. The central Mongolian forest-steppe is a combination of ecosystems, a complex of ecosystems or a landscape, but not an ecotone. The authors should avoid the use of this word and replace it.

Details of the measurements of the field capacity should be given (no mention of it in the materials and methods part).

---

## Author Comment (AC1) · 17 Dec 2020

Thank you for your valuable suggestions to improve the manuscript. We will reply to each of them below.

Specific comments:

Yes, we agree that "soil hydrology" needs to be included in the keywords. We prefer to keep "permafrost" in the keywords as well, because: although it turned out not relevant for forest regrowth, it does play an important role for forest distribution in the Mongolian forest-steppe, and we address this also several times in the paper. We will

add information on the role of permafrost in the abstract to make this clearer from the beginning. Thanks for your suggestion regarding Figure 9. We will check, how we can best include the specific factors and illustrate the strengths of their effects, e.g., by arrows with different width. We will consider this comment by including low, moderate and severe disturbance more explicitly in the Discussion section.

Technical corrections:

We will reduce the number of citations, as you proposed. We will add the missing company name of the analytical instrument (Eijkelkamp). Yes, we will add the figure numbers behind "PCA" etc. to guide the reader directly to the respective figures. Yes, sorry. We will delete textbooks from the references. The cited PhD theses cannot be avoided, because they contain relevant information that was not published in a journal. Therefore, we cannot mention this information without citing these PhD theses. We are sorry about the mistakes in the reference list and will correct it according to your comments, concerning differences in abbreviations, spelling and missing information. We will reduce the German references as much as possible. However, we cannot ignore the Russian and Mongolian references, as they provide a source of information that is not available elsewhere. We will move the climate diagram to the supplement.

---

## Author Comment (AC2) · 17 Dec 2020

Thank you for the valuable comments. We will reply to each comment below.

Specific comments:

1) We agree with your statement. It is difficult to draw conclusions from an uneven comparison without statistical prove. 2)We will add specific information in the methods to clarify this problem. The numbers are different, because we took undisturbed samples for pF curve measurements wherever possible, but high amounts of rock-fragments did not allow for taking undisturbed samples in some profiles. As this resulted in different numbers, we avoided a statistical analysis. We decided only to present a site-wise comparison with the purpose just to underline the already described significant difference in soil properties. We present the entire data set to avoid any bias. The different numbers of measurements of hydraulic conductivity and field capacity result from the in-situ measurements of hydraulic conductivity. These allowed us to generate hydraulic-conductivity data also for profiles where cylinder sampling was not possible. 3) Thanks for pointing to these difficulties. We will increase the readability of this figure: We will generate site numbers and will write those under the profiles in Figure 8. This will make it easier for the reader to compare e.g., DWIR at site 1 with DNOR at site 1. We think that a statistical analysis of this data is not feasible in this case, because of the unequal representation of the different vegetation categories and the comparably low number of measurements. In order to enable statistical treatment of the plant-available field capacities, we can in addition calculate plant-available field capacity for all profiles (based on texture, humus content and rock-fragment content), and present the results in the same way as we present the texture data in Figs. 4 and 5. This might be the most meaningful way forward, as actually, the trees do not respond directly to the different sand and silt contents that are shown in Figs. 4 and 5, but they respond to the different plant-available field capacities that result from these textural differences.

Technical comment:

Yes, some authors regard ecotones as narrow belts, but others use the term in the same way we do, i.e., for the spatial-transition character of the Mongolian forest-steppe. The reference list of our manuscript also includes several papers written by ecologists who use the term "ecotone" for the Mongolian forest-steppe (Dulamsuren et al., 2009, 2011; Sankey et al., 2006) Therefore, we prefer to keep the term. We will add details on the measurement of the pF curves, from which we obtained plant-available field capacity in the methods part.

---

## Author Response (AR3)

**Authors' response to referee #1:**

Thank you for your valuable suggestions to improve the manuscript. We will reply to each of them below.

Specific comments:

- We added "soil hydrology" to the keywords. We preferred to keep "permafrost" in the keywords as well, because: although it turned out not relevant for forest regrowth, it does play an important role for forest distribution in the Mongolian forest-steppe, and we address this also several times in the paper. We added information on the role of permafrost in the abstract to make this clearer from the beginning.
- We changed Figure 9 concerning the arrow width and added further information to improve this figure.
- We included low, moderate and severe disturbance more explicitly in the discussion section.

Technical corrections:

- We reduced the number of citations, as you proposed.
- We reduced German references, as you proposed. However, we cannot ignore the Russian and Mongolian references, as they provide a source of information that is not available elsewhere.
- We added the missing company name of the analytical instrument (Eijkelkamp).
- We added the figure number behind "PCA" etc. to guide the reader directly to the respective figures.
- We deleted textbooks from the references.
- The cited PhD theses cannot be avoided, because they contain relevant information that was not published in a journal. Therefore, we kept citing these PhD theses.
- We corrected the reference list concerning differences in abbreviations, spelling and missing information wherever we could.
- We moved the climate diagram to the supplement.

**Authors' response to referee #2:**

Thank you for the valuable comments. We will reply to each comment below.

Specific comments:

- We added specific information concerning sampling and laboratory measurements for plant-available field capacity. We explained why the number of hydraulic conductivity and plant-available field capacity measurements are unequal.
- We increased the readability of figure 8 by generating site numbers to make it easier for the reader to compare the sites.
- We added calculated plant-available field capacity using a pedo-transfer function. We analysed this data and added the information and a figure in the results, discussion and conclusions section.

Technical comment:

- We avoided the use of the term "ecotone" in the entire manuscript.
- We added details on the measurement of the pF curves, from which we obtained plant-available field capacity, in the methods part.

**Further improvements:**

- We adjusted the figure sizes to place them together with the figure captions on one page.
- We moved text within the sections to increase readability in context with the figures.
- We changed Table A1 to an editable format.
- We improved the language.